# LOCAL SEARCH GFLOWNETS

**Minsu Kim**[*] **& Taeyoung Yun**
KAIST

**Emmanuel Bengio**
Recursion

**Dinghuai Zhang**
Mila, Université de Montréal

**Yoshua Bengio**
Mila, Université de Montréal, CIFAR

**Sungsoo Ahn**
POSTECH

**Jinkyoo Park**
KAIST, Omelet

## ABSTRACT

Generative Flow Networks (GFlowNets) are amortized sampling methods that learn a distribution over discrete objects proportional to their rewards. GFlowNets exhibit a remarkable ability to generate diverse samples, yet occasionally struggle to consistently produce samples with high rewards due to over-exploration on wide sample space. This paper proposes to train GFlowNets with local search, which focuses on exploiting high-rewarded sample space to resolve this issue. Our main idea is to explore the local neighborhood via backtracking and reconstruction guided by backward and forward policies, respectively. This allows biasing the samples toward high-reward solutions, which is not possible for a typical GFlowNet solution generation scheme, which uses the forward policy to generate the solution from scratch. Extensive experiments demonstrate a remarkable performance improvement in several biochemical tasks. Source code is available: https://github.com/dbsxodud-11/ls_gfn.

## 1 INTRODUCTION

Generative Flow Networks (GFlowNets, Bengio et al., 2021) are a family of probabilistic models designed to learn reward-proportional distributions over objects, in particular compositional objects constructed from a sequence of actions, e.g., graphs or strings. It has been used in critical applications, such as molecule discovery (Li et al., 2022; Jain et al., 2023a), multi-objective optimization (Jain et al., 2022b), biological design (Jain et al., 2022a), causal modeling (Deleu et al., 2022; Atanackovic et al., 2023; Deleu et al., 2023), system job scheduling (Zhang et al., 2022a), and graph combinatorial optimization (Zhang et al., 2023b).

GFlowNets distinguish themselves by aiming to produce a diverse set of highly rewarding samples (modes) (Bengio et al., 2021), which is especially beneficial in a scientific discovery process where we need to increase the number of candidates who survive even after screening by the true oracle function. In pursuit of this objective, the use of GFlowNets emphasizes exploration to uncover novel modes that differ significantly from previously collected data points.

However, GFlowNets occasionally fall short in collecting highly rewarding experiences as they become overly fixated on exploring the diverse landscape of the vast search space during training. This tendency ultimately hinders their training efficiency, as GFlowNets heavily relies on experiential data collected by their own sampling policy (Shen et al., 2023).

**Contribution.** In this study, we introduce a novel algorithm, local search GFlowNets (LS-GFN), which is designed to enhance the training effectiveness of GFlowNets by leveraging local search in object space. LS-GFN has three iterative steps: (1) we *sample* the complete trajectories using GFlowNet trajectories; (2) we *refine* the trajectories using local search; (3) we *train* GFlowNets using revised trajectories. LS-GFN is promising, as we synergetically combine *inter-mode* global exploration and *intra-mode* local exploration. GFlowNets induce inter-mode

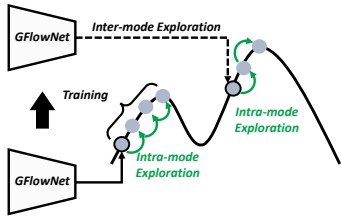

Figure 1: Strategy of LS-GFN.

---
[*]Correspondence to: Minsu Kim <min-su@kaist.ac.kr>

exploration via the iterative construction of solutions from scratch. As shown in Figure 1, local search serves as a means to facilitate intra-mode exploration.

Our extensive experiments underscore the effectiveness of the proposed exploration strategy for GFlowNets. To assess the efficacy of our method, we apply it to six well-established benchmarks encompassing molecule optimization and biological sequence design. We observe a significant improvement in the mode seeking and average reward of GFlowNets with our local search. The proposed method outperforms not only prior GFlowNet methods but also reward-maximization techniques employed by various reinforcement learning baselines as well as sampling baselines, in terms of both the number of modes discovered and the value of top-K rewards.

## 2 RELATED WORKS

**Advances and extension of GFlowNets.** A GFlowNet is a generative model that learns particle flows on a directed acyclic graph (DAG), with directed edges denoting actions and nodes signifying states of the Markov decision process (MDP). The quantity of flows it handles effectively represents the unnormalized density within the generation process. GFlowNets, when introduced initially by Bengio et al. (2021) for scientific discovery (Jain et al., 2023b), employed a flow matching condition for their temporal difference (TD)-like training scheme. This condition ensures that all states meet the requirement of having equal input and output flows. Subsequent works have further refined this objective, aiming for more stable training and improved credit assignment. Notably, Malkin et al. (2022) introduced trajectory balance which predicts the flow along complete trajectories, resembling a Monte Carlo (MC) method to achieve unbiased estimation. Madan et al. (2023) proposed subtrajectory balance, which is akin to TD($\lambda$) (Sutton, 1988), to train GFlowNets from partial trajectories. Furthermore, Zhang et al. (2023c) proposed quantile matching to better incorporate uncertainty in the reward function.

GFlowNets exhibits insightful connections with various research domains, enriching the synergy between these areas. In the study by Zhang et al. (2022b; 2023a), the connection between GFlowNets and generative models such as energy-based models (LeCun et al., 2006) and denoising diffusion probabilistic models (Ho et al., 2020) is investigated. Meanwhile, Malkin et al. (2023) shed light on the relationship between hierarchical variational inference (Ranganath et al., 2016) and GFlowNets, providing a comprehensive analysis of why GFlowNets deliver superior performance. Additionally, the works of Pan et al. (2022; 2023a;b) offer valuable insights into the integration of reinforcement learning techniques.

**Improving generalization of GFlowNets in high reward space.** In a prior attempt to overcome the low-reward exploration tendency of GFlowNets, Shen et al. (2023) suggested strategies such as prioritized replay training to target higher-reward regions, and structure-based credit assignment to identify shared structures among high-reward objects. Furthermore, they suggested a new edge flow parametrization method called SSR, which predicts edge flows as a function of pairs of states rather than of a single state. Although Shen et al. (2023) share a similar objective to our research, we distinguish ourselves by employing a local search approach to steer GFlowNet towards the exploration of highly rewarding regions.

## 3 PRELIMINARIES

In this section, we introduce the foundational concepts underpinning GFlowNets, a novel generative model tailored for compositional objects denoted as $x \in \mathcal{X}$. We follow the notation from Bengio et al. (2023). GFlowNets follow a trajectory-based generative process, using discrete *actions* to iteratively modify a *state* which represents a partially constructed object. This can be described by a directed acyclic graph (DAG), $G = (\mathcal{S}, \mathcal{A})$, where $\mathcal{S}$ is a finite set of all possible states, and $\mathcal{A}$ is a subset of $\mathcal{S} \times \mathcal{S}$, representing directed edges. Within this framework, we define the *children* of state $s \in \mathcal{S}$ as the set of states connected by edges whose head is $s$, and the *parents* of state $s$ as the set of states connected by edges whose tail is $s$.

We define a *complete trajectory* $\tau = (s_0 \rightarrow \ldots \rightarrow s_n) \in \mathcal{T}$ from the initial state $s_0$ to terminal state $s_n = x \in \mathcal{X}$. We define *trajectory flow* as $F(\tau) : \mathcal{T} \rightarrow \mathbb{R}_{\geq 0}$, which represents the unnormalized density function of $\tau \in \mathcal{T}$. We define *state flow* as the total amount of unnormalized probability

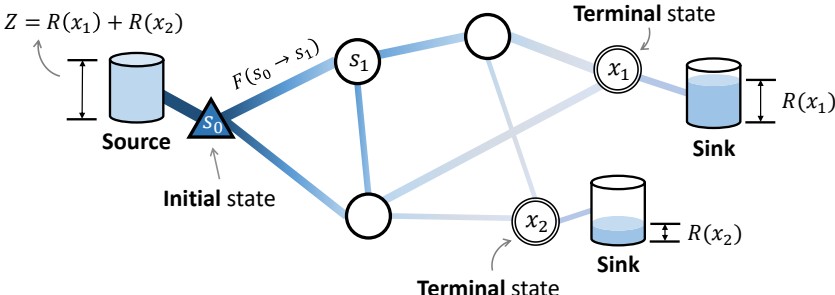

Figure 2: Illustration of a GFlowNet in a toy environment with two objects $x_1, x_2 \in \mathcal{X}$. $Z = R(x_1) + R(x_2)$ is the total amount of flow in the source, and the sink is the storage for the flow of the terminal state $x_1$ and $x_2$. The generative probability is: $p(x_1) = R(x_1) / (R(x_1) + R(x_2))$, and $p(x_2) = R(x_2) / (R(x_1) + R(x_2))$.

flowing though state $s$: $F(s) = \sum_{\tau \in \mathcal{T}: s \in \tau} F(\tau)$, and *edge flow* as the total amount of unnormalized probability flowing through edge $s \to s'$: $F(s \to s') = \sum_{\tau \in \mathcal{T}: (s \to s') \in \tau} F(\tau)$.

We define the *trajectory reward* $R(\tau)$ as the reward of the terminal state of the trajectory $R(\tau = (s_0 \to \ldots \to s_n = x)) = R(x)$, i.e., the reward is determined only by the terminal state, and intermediate states do not contribute to the reward values.

We define the *forward policy* to model the forward transition probability $P_F(s'|s)$ from $s$ to its child $s'$. Similarly, we also consider the *backward policy* $P_B(s|s')$ for the backward transition $s' \dashrightarrow s$, where $s$ is a parent of $s'$.

$P_F$ and $P_B$ are related to the Markovian flow $F$ as follows:

$$P_F(s'|s) = \frac{F(s \to s')}{F(s)}, \quad P_B(s|s') = \frac{F(s \to s')}{F(s')}$$

The marginal likelihood of sampling $x \in \mathcal{X}$ can be derived as $P_F^\top(x) = \sum_{\tau \in \mathcal{T}: \tau \to x} P_F(\tau)$ where $\tau \to x$ denotes a complete trajectory $\tau$ that terminates at $x$. The ultimate objective of GFlowNets is to match the marginal likelihood with the reward function, $P_F^\top(x) \propto R(x)$. $P_F^\top$ is also called terminating probability. See Figure 2 for a conceptual understanding of GFlowNets.

**Trajectory balance.** The trajectory balance algorithm is one of the training methods that can achieve $P_F^\top(x) \propto R(x)$. The trajectory balance loss $L_{\text{TB}}$ works by training three models, a learnable scalar of initial state flow $Z_\theta \approx F(s_0) = \sum_{\tau \in \mathcal{T}} F(\tau)$, a forward policy $P_F(s_{t+1}|s_t; \theta)$, and a backward policy $P_B(s_t|s_{t+1}; \theta)$ to minimize the following objective:

$$\mathcal{L}_{\text{TB}}(\tau; \theta) = \left( \log \frac{Z_\theta \prod_{t=1}^n P_F(s_t|s_{t-1}; \theta)}{R(x) \prod_{t=1}^n P_B(s_{t-1}|s_t; \theta)} \right)^2 \tag{1}$$

**Replay training of GFlowNets.** One of the interesting advantages of GFlowNets is that they can be trained in an off-policy or offline manner (Bengio et al., 2021). To this end, prior methods often rely on training with replay buffers, which iterate two stages. (1) collect data $\mathcal{D} = \{\tau_1, \ldots, \tau_M\}$ by using the GFlowNet's forward policy $P_F$ or an exploratory policy, and (2) minimize the loss computed from samples from the replay buffer $\mathcal{D}$. This training process leverages the generalization capability of the flow model to make good predictions on unseen trajectories, which is critical since visiting the entire trajectory space of $\mathcal{T}$ is intractable. This generalization capability highly relies on the quality of the training dataset $\mathcal{D}$. This study investigates how to make high-quality replay buffers $\mathcal{D}$ using a local search method during the sampling step.

## 4 LOCAL SEARCH GFLOWNETS (LS-GFN)

**Overview.** Our method is a simple augmentation (**Step B**) of existing training algorithms for GFlowNets (**Step A**, and **Step C**). Our local search in **Step B** refines the candidate samples by

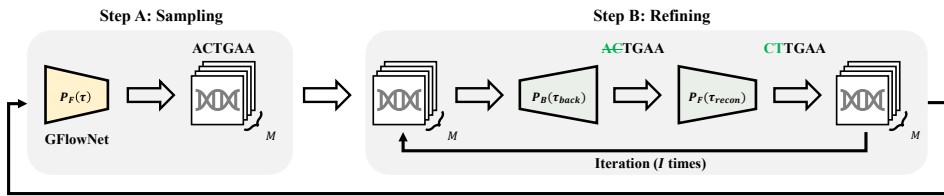

Figure 3: Illustration of Local Search GFlowNet (LS-GFN) algorithm.

partially backtracking the complete trajectory using the backward policy $P_B$ and then reconstructing it with the forward policy $P_F$. This procedure is done multiple times to iteratively refine and construct highly rewarding samples.

Our algorithm trains $P_F$ and $P_B$ by repeating these three steps (see also Fig. 3):

**Step A.** We *sample* a set of trajectories $\{\tau_1, ..., \tau_M\}$ using $P_F(\tau)$.

**Step B.** We *refine* the $M$ trajectories in parallel for $I$ iterations. For each iteration, we generate $\{\tau'_1, \cdots \tau'_M\}$ from $\{\tau_1, \cdots \tau_M\}$ using a local search by using $P_B$'s destroying and $P_F$'s backtracking, and add $\{\tau'_1, \cdots \tau'_M\}$ into the training dataset $\mathcal{D}$. Then, we choose whether to accept $\tau_m \leftarrow \tau'_m$ (i.e. make transition) or reject $\tau_m \leftarrow \tau_m$ (i.e. make staying) with filtering rules for $m = 1, \cdots, M$.

**Step C.** We *train* the GFlowNet by using training dataset $\mathcal{D}$. We use reward prioritized sampling over $\mathcal{D}$ and use the sampled trajectories to minimize a GFlowNet loss function such as trajectory balance.

## 4.1 STEP A: SAMPLING

We construct a complete trajectory $\tau$ through a sequential process of generating actions from scratch by using forward policy $P_F(\tau)$. This approach enables global exploration over different modes. It is worth noting that within the GFlowNet literature, various techniques have been proposed to use $P_F(\tau)$ for exploration (Pan et al., 2022; Rector-Brooks et al., 2023). In this work, we employ the $\epsilon$-noisy method. It selects a random action with probability $\epsilon$ and follows $P_F(s'|s)$ to sample the action with probability $1 - \epsilon$.

## 4.2 STEP B: REFINING

After completing **Step A**, we have an initial candidate set of samples $\tau_1, \tau_2, \cdots, \tau_M$. In **Step B**, we make $I$ iterations of local search for $M$ candidate trajectories in parallel.

Taking a representative among the $M$ candidate trajectories, let $\tau = (s_0 \rightarrow \ldots \rightarrow s_n = x)$. Inspired by Zhang et al. (2022b), we backtrack $K$-step from complete trajectory into partial trajectory using $P_B$. Subsequently, employing $P_F$, we sample a $K$-step forward trajectory to *reconstruct* a complete trajectory from the partial trajectory.

$$\tau_{\text{back}} = \left( x = s_n \dashrightarrow \ldots \dashrightarrow s'_{n-K} \right), \quad \tau_{\text{recon}} = \left( s'_{n-K} \rightarrow \ldots \rightarrow s'_n = x' \right) \tag{2}$$

Note the $\dashrightarrow$ stands for a backward transition from one state to its parent state. We call the local search refined trajectory $\tau'$:

$$\tau' = (s_0 \rightarrow \cdots \rightarrow \underbrace{s'_{n-K} \rightarrow \cdots \rightarrow s'_n = x'}_{\text{recon}}) \tag{3}$$

We define the transition probability $q(\tau'|\tau)$ along with its reverse counterpart $q(\tau|\tau')$ as follows:

$$q(\tau'|\tau) = P_B(\tau_{\text{back}}|x)P_F(\tau_{\text{recon}}), \quad q(\tau|\tau') = P_B(\tau_{\text{recon}}|x')P_F(\tau_{\text{destroy}}) \tag{4}$$

We now need to determine whether to accept or reject $\tau'$ from $q(\tau'|\tau)$. We present two filtering strategies, one deterministic and the other stochastic.

**Deterministic Filtering.** We accept $\tau'$ with following probability:

$$A(\tau, \tau') = 1_{\{R(\tau') > R(\tau)\}} \tag{5}$$

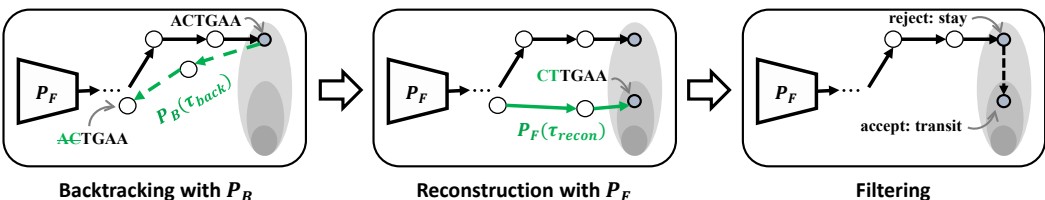

Figure 4: Illustration of the 3-step refinement process of LS-GFN.

**Stochastic Filtering.** We accept $\tau'$ using the back-and-forth Metropolis-Hastings (Hastings, 1970; Zhang et al., 2022b) acceptance probability:

$$A\left(\tau, \tau'\right) = \min\left[1, \frac{R(\tau')}{R(\tau)} \frac{q(\tau'|\tau)}{q(\tau|\tau')}\right] \qquad (6)$$

Deterministic filtering offers advantages in the context of greedy local search when aiming to maximize the rewards of candidate samples. In contrast, stochastic filtering can be viewed as a post-processing sampling method within Markov chain Monte Carlo (MCMC) that helps maintain the sampling objective of GFlowNets, where we seek to generate samples from the distribution $p(x) \propto R(x)$ while promoting diversity. In this work, we use deterministic filtering as a default setting but closely analyze its pros and cons in Appendix B.2. Our overall process of refinement, including backtracking, reconstruction, and filtering, is illustrated in Fig. 4.

Note that we gather both accepted and rejected trajectories and compile them into a training dataset $\mathcal{D}$. To ensure that highly rewarded trajectories from $\mathcal{D}$ receive priority during training, we use a reward-based prioritized replay training (PRT) method (Shen et al., 2023) . This approach increases the likelihood of using accepted trajectories within the training process.

### 4.3 STEP C: TRAINING

In this work, we use the TB objective function as the default objective for training on a dataset $\mathcal{D}$. In TB, we train three models, a model of initial state flow $Z_\theta$, a forward policy $P_F(s_{t+1}|s_t; \theta)$, and a backward policy $P_B(s_t|s_{t+1}; \theta)$ as follows:

$$\mathcal{L}(\theta; \mathcal{D}) = \mathbb{E}_{P_{\mathcal{D}}(\tau)}\left[\left(\log \frac{Z_\theta \prod_{t=1}^{n} P_F\left(s_t|s_{t-1}; \theta\right)}{R(x) \prod_{t=1}^{n} P_B\left(s_{t-1}|s_t; \theta\right)}\right)^2\right]. \qquad (7)$$

Note that the transition probability $q(\tau'|\tau)$ in Equation (3) is defined with $P_F(s_{t+1}|s_t; \theta)$ and $P_B(s_t|s_{t+1}; \theta)$ where this means the local search capability also evolves with GFlowNets and improves throughout the training process.

---

**Algorithm 1** Local Search GFlowNet (LS-GFN)

---

1: Set $\mathcal{D} \leftarrow \emptyset$             ▷ *Initialize training dataset.*
2: **for** $t = 1, \ldots, T$ **do**            ▷ *Iteration of training rounds*
3:   Sample $\tau_1, ..., \tau_M \sim P_F(\tau; \theta)$       ▷ **Step A:** *Sampling*
4:   **for** $i = 1, \ldots, I$ **do**          ▷ **Step B:** *Refining*
5:     **for** $m = 1, \ldots, M$ **do**
6:       Propose $\tau'_m \sim q(\cdot|\tau_m; \theta)$ by Equation (4).
7:       Update $\mathcal{D} \leftarrow \mathcal{D} \cup \{\tau'_m\}$
8:       Accept $(\tau_m \leftarrow \tau'_m)$ or reject $(\tau_m \leftarrow \tau_m)$ by Equation (5).
9:     **end for**
10:   **end for**
11:   Use the Adam optimizer to achieve: $\theta \leftarrow \arg\min \mathcal{L}(\theta; \mathcal{D})$.    ▷ **Step C:** *Training*
12: **end for**

---

We use reward-based prioritized replay training (PRT) (Shen et al., 2023), where we set $P_\mathcal{D}(\tau)$ to sample a batch of trajectories $\mathcal{B} = \{\tau_1, \ldots, \tau_M\}$ where 50% of the $\mathcal{B}$ is sampled from the above 90th percentile of the $\mathcal{D}$ and the 50% of the $\mathcal{B}$ is sampled from the below 90th percentile of the $\mathcal{D}$.

Our method can similarly accommodate various other objective functions, such as DB, and SubTB. See Algorithm 1 for the detailed pseudocode of our method.

## 5 EXPERIMENTS

We present our experimental results on 6 biochemical tasks, including molecule optimization and biological sequence design. In these settings, generating diverse samples with relatively high rewards is crucial for robustness to proxy misspecification (Bengio et al., 2023). To this end, we measure the accuracy of GFlowNets using the relative gap to the target reward distribution following Shen et al. (2023). We also measure the number of modes discovered by GFlowNets.

### 5.1 TASK DESCRIPTION

Let $\mathcal{X}$ be the set of all objects that can be generated (i.e., the terminal state space), and $\mathcal{T}$ be the complete trajectory space which consists of all possible trajectories that can incrementally construct any $x \in \mathcal{X}$. As different trajectories $\tau_1, \ldots, \tau_N \in \mathcal{T}$ can represent identical $x \in \mathcal{X}$, $|\mathcal{T}| \geq |\mathcal{X}|$.

We consider two molecule optimization and four biological sequence design tasks:

**QM9.** Our goal is to generate a small molecule graph. We have 12 building blocks with 2 stems and generate a molecule with 5 blocks. Our objective is to maximize the HOMO-LUMO gap, which is obtained via a pre-trained MXMNet (Zhang et al., 2020) proxy.

**sEH.** Our goal is to generate binders of the sEH protein. We have 18 building blocks with 2 stems and generate a molecule with 6 blocks. Our objective is to maximize binding affinity to the protein provided by the pre-trained proxy model provided by (Bengio et al., 2021).

**TFBind8.** Our goal is to generate a string of length 8 of nucleotides. Though an autoregressive MDP is conventionally used for strings, we use a prepend-append MDP (PA-MDP) (Shen et al., 2023), in which the action involves either adding one token to the beginning or the end of a partial sequence. The reward is a DNA binding affinity to a human transcription factor (Trabucco et al., 2022).

**RNA-Binding.** Our goal is to generate a string of 14 nucleobases. We consider the PA-MDP to generate strings. Our objective is to maximize the binding affinity to the target transcription factor. We present three different target transcriptions, L14-RNA1, L14-RNA2, and L14-RNA3, introduced by Sinai et al. (2020).

### 5.2 BASELINES

We consider prior GFlowNet (GFN) methods and reward-maximization methods as our baselines. Prior GFN methods include detailed balance (DB, Bengio et al., 2023), maximum entropy GFN (MaxEnt, Malkin et al., 2022), trajectory balance (TB, Malkin et al., 2022), sub-trajectory balance (SubTB, Madan et al., 2023), and substructure-guided trajectory balance (GTB, Shen et al., 2023). For reward-maximization methods, we consider Markov Molecular Sampling (MARS, Xie et al., 2020), which is a sampling-based method known to work well in the molecule domain, and RL-based methods which include advantage actor-critic (A2C) with entropy regularization (Mnih et al., 2016), Soft Q-Learning (SQL, Haarnoja et al., 2018), and proximal policy optimization (PPO, Schulman et al., 2017).

### 5.3 IMPLEMENTATIONS AND HYPERPARAMETERS

For GFN implementations, we strictly follow implementations from Shen et al. (2023) and re-implement only non-existing methods by ourselves. For all GFN models, we apply prioritized replay training (PRT) and relative edge flow policy parametrization mapping (SSR) from Shen et al. (2023). We run experiments with $T = 2,000$ training rounds for QM9, sEH, and TFBind8 and $T = 5,000$ training rounds for RNA-binding tasks.

Table 1: Accuracy of GFlowNets. Mean and standard deviation from 3 random seeds are reported.

| Method | QM9 (↑) | sEH (↑) | TFBind8 (↑) | L14-RNA1 (↑) | L14-RNA2 (↑) | L14-RNA3 (↑) |
|---|---|---|---|---|---|---|
| DB | 93.16 ± 0.94 | 95.26 ± 0.37 | 77.64 ± 0.70 | 28.25 ± 0.54 | 16.99 ± 0.15 | 17.27 ± 0.21 |
| DB + LS-GFN | 95.41 ± 1.94 | 93.77 ± 0.48 | 75.59 ± 0.09 | 29.86 ± 0.24 | 18.19 ± 0.31 | 17.72 ± 0.03 |
| MaxEnt | 96.95 ± 0.44 | 100.00 ± 0.00 | 84.64 ± 0.63 | 33.53 ± 0.19 | 21.80 ± 0.26 | 32.49 ± 1.59 |
| MaxEnt + LS-GFN | 100.00 ± 0.00 | 100.00 ± 0.00 | 97.67 ± 1.14 | 88.04 ± 1.94 | 56.93 ± 1.05 | 74.28 ± 3.71 |
| SubTB (0.9) | 93.49 ± 0.62 | 98.98 ± 0.19 | 76.53 ± 1.08 | 29.38 ± 0.32 | 28.18 ± 0.14 | 18.77 ± 0.27 |
| SubTB (0.9) + LS-GFN | 100.00 ± 0.00 | 100.00 ± 0.00 | 76.54 ± 0.55 | 41.16 ± 0.41 | 25.01 ± 0.31 | 21.24 ± 0.12 |
| TB | 97.84 ± 0.63 | 100.00 ± 0.00 | 85.63 ± 0.35 | 33.47 ± 0.37 | 21.88 ± 0.35 | 32.70 ± 0.59 |
| TB + LS-GFN | 100.00 ± 0.00 | 100.00 ± 0.00 | 97.05 ± 0.58 | 87.28 ± 3.25 | 56.63 ± 0.56 | 75.75 ± 3.10 |

Figure 5: Accuracy of GFlowNet on various tasks. Ours stands for TB + LS-GFN.

To ensure fairness in sample efficiency across all baselines, we maintain a consistent reward evaluation budget for each task. This budget denoted as $B$, is determined by the number of candidate samples per training round ($M$), and the number of local search revisions ($I$) resulting in $B = M \times (I + 1) = 32$ for all baselines. for LS-GFN, we set $M = 4$, and $I = 7$ as default. We provide a detailed description of the hyperparameters in Appendix A.2.

## 5.4 EVALUATING THE ACCURACY OF GFLOWNETS

We first evaluate how well our method matches the target reward distribution. As suggested in Shen et al. (2023), we measure the accuracy of training GFlowNet $p(x; \theta)$ by using a relative error between the sample mean of $R(x)$ under the learned distribution $p(x; \theta)$ and the expected value of $R(x)$ given the target distribution $p^*(x) = R(x) / \sum_{x \in \mathcal{X}} R(x)$:

$$\text{Acc}\left(p\left(x; \theta\right)\right) = 100 \times \min\left(\frac{\mathbb{E}_{p(x;\theta)}\left[R\left(x\right)\right]}{\mathbb{E}_{p^*(x)}\left[R\left(x\right)\right]}, 1\right),$$

For all experiments, we report the performance with three different random seeds. We provide details of our experiments in Appendix A.1.

Table 1 presents the results of our method when integrated with different GFN training objectives. Note that our local search mechanism is orthogonal to training methods, so we can plug our method into various objectives. As shown in the table, our method outperforms baselines and matches the target distribution in most cases. This highlights the effectiveness of local search guided by GFN policies on finding high-quality samples.

Figure 5 shows the performance of our method and prior GFN baselines across training. We only plot results of prior GFN methods without local search and our method integrated with TB for clear visualization. For monitoring, we collect 128 on-policy samples every 10 training rounds and accumulate them, following Shen et al. (2023). Note that samples for computing relative error from the target mean have never been used for training. As shown in the figure, our method converges to the target mean faster than any other baselines.

Table 2: The number of discovered modes. Mean and standard deviation from 3 random seeds are reported

| Method | QM9 ($\uparrow$) | sEH ($\uparrow$) | TFBind8 ($\uparrow$) | L14-RNA1 ($\uparrow$) | L14-RNA2 ($\uparrow$) | L14-RNA3 ($\uparrow$) |
|---|---|---|---|---|---|---|
| DB | 635 ± 5 | 217 ± 11 | 304 ± 5 | 5 ± 0 | 4 ± 0 | 1 ± 0 |
| DB + LS-GFN | 745 ± 5 | 326 ± 13 | 317 ± 0 | 11 ± 3 | 13 ± 1 | 3 ± 0 |
| MaxEnt | 701 ± 10 | 676 ± 37 | 316 ± 3 | 10 ± 1 | 8 ± 2 | 7 ± 3 |
| MaxEnt + LS-GFN | 793 ± 3 | 4831 ± 148 | 317 ± 2 | 33 ± 2 | 31 ± 1 | 19 ± 0 |
| SubTB (0.9) | 665 ± 8 | 336 ± 28 | 309 ± 3 | 6 ± 0 | 5 ± 1 | 4 ± 1 |
| SubTB (0.9) + LS-GFN | 787 ± 2 | 2434 ± 60 | 314 ± 2 | 16 ± 4 | 13 ± 0 | 7 ± 0 |
| TB | 699 ± 14 | 706 ± 126 | 320 ± 3 | 10 ± 4 | 6 ± 0 | 6 ± 1 |
| TB + LS-GFN | 793 ± 4 | 5228 ± 141 | 316 ± 0 | 32 ± 4 | 27 ± 1 | 18 ± 0 |

Figure 6: Number of modes discovered over training. Ours stands for TB + LS-GFN.

**Note**: We conducted a local search only for training, not at the inference phase for fair comparison. For every inference-aware metric, such as the relative error metric, we compare our method with other GFN baselines without the local search refining process.

## 5.5 Evaluating the Number of Modes Discovered

In this experiment, we systematically assess our training process's ability to uncover numerous distinctive modes. In biochemical tasks, modes are defined as high-scoring samples that exceed a specified reward threshold, and are distinctly separated based on a predefined similarity constraint. To achieve this, we evaluate both the reward magnitude and the diversity of generated samples. Detailed statistics regarding these modes can be found in Appendix B.4. To ensure the reliability of our results, we report performance across all experiments using three random seeds.

In Table 2, we present the outcomes of incorporating our method into various GFN training objectives. We see that our approach exhibits remarkable performance in mode diversity when compared to previous GFN techniques. This underscores the efficacy of our local search mechanism in facilitating exploration within intra-mode regions during training. Notably, our method showcases the most substantial improvements in RNA-binding tasks, where objects are comparatively longer than in other tasks. Complementing these findings, Figure 6 visually represents the progression of mode discovery throughout training. Our method not only identifies the highest number of modes among the compared techniques but also stands out for its accelerated mode detection, underscoring its efficiency. This is relatively surprising, since one common downside of training on higher rewards is to make the model greedier and *less* diverse (Jain et al., 2023c).

## 5.6 Comparison with Reward Maximization Methods

We evaluate our method against established techniques, including reinforcement learning baselines (PPO, A2C, SQL) and a sampling baseline (MARS). We use three metrics: number of modes discovered, mean rewards of the top 100 scoring samples out of evaluation samples accumulated across

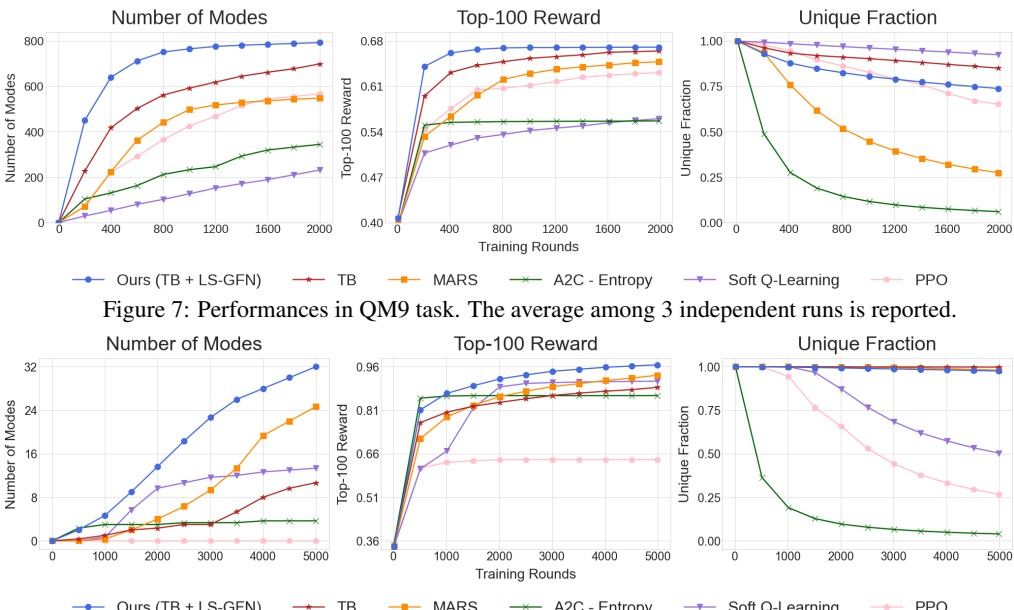

Figure 7: Performances in QM9 task. The average among 3 independent runs is reported.

Figure 8: Performances in L14-RNA1 task. The average among 3 independent runs is reported.

training, and sample uniqueness. Sample uniqueness is maximized at 1.0 when all samples are distinct, while it will be zero when all samples are identical.

As shown in Figures 7 and 8, our method surpasses reward-maximization methods in terms of mode-seeking capabilities. Reward maximization methods can lead to a high fraction of duplicated samples, falling into non-diverse local optima. Our method consistently surpasses existing techniques in terms of the number of modes identified, which is only possible when both strong exploration *and* exploitation are achieved by the model.

We interpret these results by recalling the importance of the structure of *both* trajectory space ($\mathcal{T}$) and object space ($\mathcal{X}$). Some inefficiencies in reinforcement learning (RL) arise from the failure to account for symmetries, wherein multiple trajectories can lead to the generation of identical samples. GFlowNets, which make use of this symmetry in their training objective, may very well waste less time visiting the same state from different paths, since they are trained to know they are the same outcome. See Appendix B.1 for detailed results on the other four tasks.

## 5.7 ADDITIONAL EXPERIMENTS

**Comparison between deterministic filtering and stochastic filtering.** See Appendix B.2.

**Experiments for hyperparameter $I$.** We did experiments for the hyperparameter $I$ we introduced, which is the number of revisions with local search; see Appendix B.3 for details.

**Experiments for the number of modes metric.** We investigated different ways of counting modes and closely compared LS-GFN with other algorithms; see Appendix B.4.

**Experiment for acceptance rate.** We measured the acceptance rate $A(\tau, \tau')$ during training, reflecting the success of the local search compared to GFlowNet's sampling. We observed an interesting phenomenon: the rate is fairly steady, signifying consistent evolution between GFlowNet ($P_F(\tau; \theta)$) and the local search (i.e., $P_B(\tau_{\text{destroy}}; \theta)$ and $P_F(\tau_{\text{recon}}; \theta)$); see Appendix B.6.

## 6 DISCUSSION

In this paper, we proposed a novel algorithm: Local Search GFlowNet (LS-GFN). We found that LS-GFN has the fastest mode mixing capability among GFlowNet baselines and RL baselines and has better sampling quality than GFlowNets. Our method had been consistently applied to exist-

ing GFlowNets algorithms with simple modifications. These results suggested that combining the inter-mode exploration capabilities of GFlowNets and intra-mode exploration through local search methods is a powerful paradigm.

**Limitation and Future Works.** A limitation of LS-GFN lies in the potential impact of the quality of the backward policy on its performance, particularly when the acceptance rate of the local search becomes excessively low. One immediate remedy is to introduce an exploratory element into the backward policy, utilizing techniques like $\epsilon$-greedy or even employing a uniform distribution to foster exploration within the local search. A promising avenue for future research could involve fine-tuning backward policy to enhance the local search's acceptance rate.

## ACKNOWLEDGEMENT

We thank Nikolay Malkin, Hyeonah Kim, Sanghyeok Choi, Jarrid Rector-Brooks, Chenghao Liu, Ling Pan, and Max W. Shen for their valuable input and feedback on this project.

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

## A  EXPERIMENTAL SETTING

### A.1  DETAILED IMPLEMENTATION

For the GFlowNets policy model, we use an MLP architecture with relative edge flow parameterization (SSR) suggested in Shen et al. (2023). Given a pair of states $(s, s')$, we encode each state into a one-hot encoding vector and concatenate them to pass as an input of the forward/backward policy network. The number of layers and hidden units varies across different tasks, which is listed in Table 3. We use the same architecture with different parameters to model forward and backward policies. We initialize $\log Z_\theta$ to 5.0. Following Shen et al. (2023), we clip gradient norms to a maximum of 10.0 and policy logit predictions to a minimum of -50.0 and a maximum of 50.0. To implement DB and SubTB, which require state flow predictions, we find that introducing a separate neural network for mapping $f_\theta(s) : \mathcal{S} \to \mathbb{R}^+$ is more useful than SSR, $f_\theta(s) = \sum_{s' \in \text{child}(s)} f_\theta(s, s')$. Please refer Figure 9.

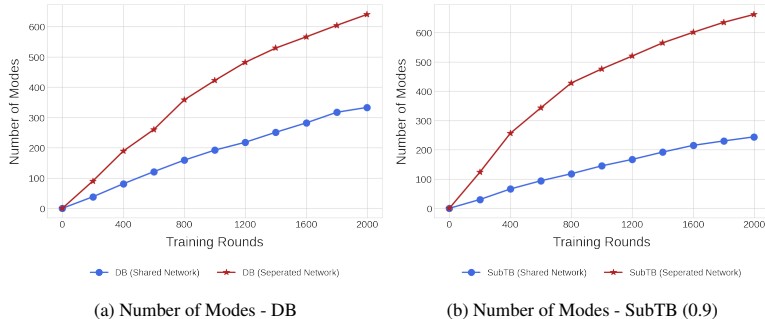

(a) Number of Modes - DB                     (b) Number of Modes - SubTB (0.9)

Figure 9: Experiments on the different parametrization of state flow in DB and SubTB.

### A.2  HYPERPARAMETERS

For hyperparameters of GFlowNets, we do not change the initial setting proposed by Shen et al. (2023). For all tasks, we use ADAM (Kingma & Ba, 2015) optimizer with learning rate $1 \times 10^{-2}$ for $\log Z_\theta$, $1 \times 10^{-4}$ for forward and backward policy. We use different reward exponent $\beta$ to make $p(x; \theta) \propto R^\beta(x)$ and reward normalization constant suggested in Shen et al. (2023) except for the RNA task, which is newly suggested by us. For the RNA task, we use a reward exponent of 8 and scale the reward to a maximum of 10.

Table 3: GFlowNet hyperparameters for various tasks

| Tasks | Number of Layers | Hidden Units | Reward Exponent ($\beta$) | Training Rounds ($T$) |
|---|---|---|---|---|
| QM9 | 2 | 1024 | 5 | 2,000 |
| sEH | 2 | 1024 | 6 | 2,000 |
| TFBind8 | 2 | 128 | 3 | 2,000 |
| RNA-binding | 2 | 128 | 8 | 5,000 |

For LS-GFN, we have set the number of candidate samples as $M = 4$ and the local search interaction to $I = 7$ as default values. In contrast, other GFN models without local search employ a default value of $M = 32$ to ensure a fair comparison of sample efficiency.

### A.3  HYPERPARAMETER TUNING FOR RL BASELINES

To implement RL baselines, we also employ the same MLP architecture used in GFlowNet baselines. We find an optimal hyperparameter by grid search on the QM9 task in terms of the number of modes. For A2C with entropy regularization, we separate parameters for actor and critic networks and use a learning rate of $1 \times 10^{-4}$ selected from $\{1 \times 10^{-5}, 1 \times 10^{-4}, 1 \times 10^{-4}, 5 \times 10^{-3}, 1 \times 10^{-3}\}$ with entropy regularization coefficient $1 \times 10^{-2}$ selected from $\{1 \times 10^{-4}, 1 \times 10^{-3}, 1 \times 10^{-2}\}$. For Soft Q-Learning, we use learning rate of $1 \times 10^{-4}$ selected from $\{1 \times 10^{-5}, 1 \times 10^{-4}, 1 \times 10^{-4}, 5 \times 10^{-3}, 1 \times 10^{-3}\}$. For PPO, we employ entropy regularization term and use a learning rate of $1 \times 10^{-4}$ selected from $\{1 \times 10^{-5}, 1 \times 10^{-4}, 1 \times 10^{-4}, 5 \times 10^{-3}, 1 \times 10^{-3}\}$ with entropy regularization coefficient $1 \times 10^{-2}$ selected from $\{1 \times 10^{-4}, 1 \times 10^{-3}, 1 \times 10^{-2}\}$.

# B ADDITIONAL EXPERIMENTS

## B.1 CLOSER COMPARISON WITH RL BASELINES

We also assess our approach against RL baselines across four additional tasks, as detailed in Chapter 5.6. In Figures 10, 11, 12, and 13, we present the comprehensive results. These findings demonstrate that our method outperforms RL baselines, particularly in the detection of diverse modes. While most RL methods yield a subpar unique fraction by producing duplicated samples concentrated in narrow, highly rewarded regions, our approach excels in seeking remarkable modes, resulting in a wide variety of highly rewarded samples.

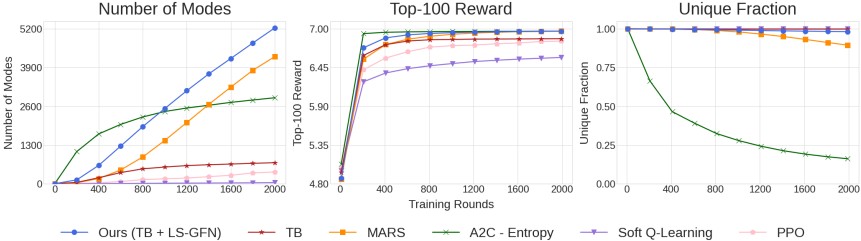

Figure 10: The sEH task.

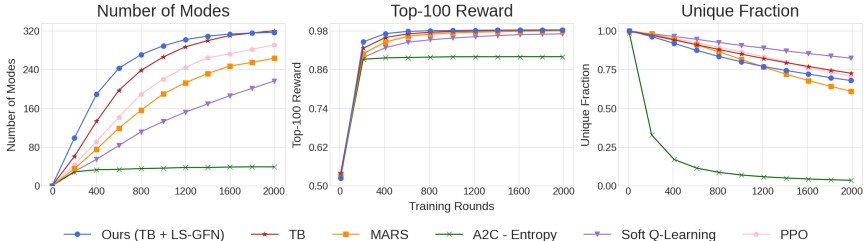

Figure 11: The TFbind8 task.

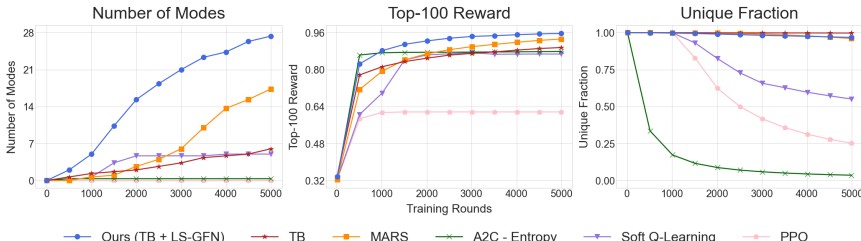

Figure 12: The L14_RNA2 task.

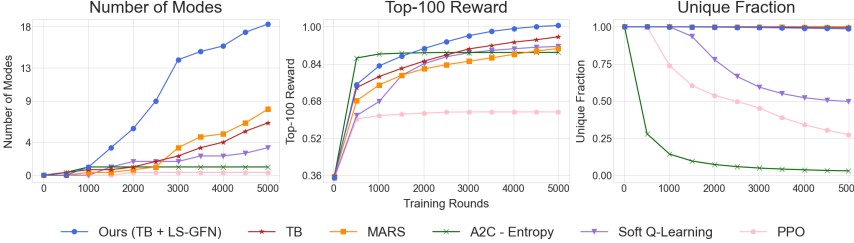

Figure 13: The L14_RNA3 task.

### B.2 CLOSER COMPARISON BETWEEN DETERMINISTIC FILTERING AND STOCHASTIC FILTERING

We also compare the different filtering strategies we proposed in the methodology section. We conduct experiments on the QM9, sEH, and TFbind8 tasks with TB as an underlying GFN training method. For evaluation, we generate 2048 samples from the trained model. Experiment results are reported in Table 4. As depicted in Table 4, the stochastic filtering strategy yields a wider range of solutions, emphasizing diversity, whereas the deterministic strategy places greater emphasis on maximizing high-scoring rewards. Consequently, these two filtering strategies can be selected based on distinct objectives or purposes.

Table 4: Analysis on Different Filtering Strategies

| Task | Filtering Strategy | Accuracy | Top 100 Reward | Top 100 Diversity | Uniq. Fraction |
|------|-------------------|----------|----------------|-------------------|----------------|
| QM9 | *Stochastic* | $100.00 \pm 0.00$ | $0.59 \pm 0.01$ | $0.43 \pm 0.00$ | $0.97 \pm 0.00$ |
| | *Deterministic* | $100.00 \pm 0.00$ | $0.61 \pm 0.02$ | $0.42 \pm 0.00$ | $0.96 \pm 0.01$ |
| sEH | *Stochastic* | $100.00 \pm 0.00$ | $6.84 \pm 0.01$ | $0.30 \pm 0.00$ | $1.00 \pm 0.00$ |
| | *Deterministic* | $100.00 \pm 0.00$ | $6.87 \pm 0.01$ | $0.29 \pm 0.01$ | $1.00 \pm 0.00$ |
| TFbind8 | *Stochastic* | $99.23 \pm 1.09$ | $0.97 \pm 0.00$ | $1.98 \pm 0.02$ | $0.96 \pm 0.00$ |
| | *Deterministic* | $100.00 \pm 0.00$ | $0.97 \pm 0.00$ | $1.94 \pm 0.03$ | $0.95 \pm 0.00$ |

### B.3 ABLATION STUDY OF $I$ AND $M$

We investigate the effect of the number of revision steps on reward and diversity. When we set the number of revision steps as 0, it is a typical GFN method. When we set the number of revision steps as a batch size, we generate a single sample and apply local search repeatedly. We conduct experiments on the QM9 task with TB as an underlying GFN training method. Table 5 presents the performance across different numbers of revision steps. As shown in the table, we confirm that the mean of the top 100 rewards consistently increases as the number of revision steps increases due to strong local exploration, while the unique fraction of samples gradually decreases.

Table 5: Effect of the number of revision steps on Reward and Diversity

| $I$ | $M$ | Num. Modes | Accuracy | Top 100 Reward | Top 100 Diversity | Uniq. Fraction |
|-----|-----|------------|----------|----------------|-------------------|----------------|
| 0 | 32 | $699 \pm 14$ | $98.46 \pm 2.17$ | $0.57 \pm 0.01$ | $0.43 \pm 0.00$ | $0.98 \pm 0.00$ |
| 1 | 16 | $752 \pm 7$ | $99.85 \pm 0.16$ | $0.57 \pm 0.01$ | $0.43 \pm 0.00$ | $0.98 \pm 0.00$ |
| 3 | 8 | $781 \pm 5$ | $100.00 \pm 0.00$ | $0.59 \pm 0.01$ | $0.42 \pm 0.00$ | $0.97 \pm 0.00$ |
| 7 | 4 | $793 \pm 4$ | $100.00 \pm 0.00$ | $0.60 \pm 0.00$ | $0.43 \pm 0.00$ | $0.97 \pm 0.00$ |
| 15 | 2 | $800 \pm 3$ | $100.00 \pm 0.00$ | $0.61 \pm 0.02$ | $0.42 \pm 0.00$ | $0.96 \pm 0.01$ |
| 31 | 1 | $793 \pm 1$ | $100.00 \pm 0.00$ | $0.62 \pm 0.01$ | $0.42 \pm 0.00$ | $0.95 \pm 0.01$ |

### B.4 EXPERIMENTS ON SEVERAL NUMBER OF MODES METRIC

How to define mode is not a trivial problem. All samples whose reward is above a certain threshold cannot be considered as modes. Therefore, we conduct experiments on several different metrics for defining modes.

First, for molecule optimization tasks, we use the Tanimoto diversity metric. We define mode as follows. For all samples whose reward is above a certain threshold level, we only accept samples that are far away from previously accepted modes in terms of diversity metric.

For biological sequence design tasks, we define mode as a local optimum among its intermediate neighborhoods. We can define the neighborhood as $n-$ hamming ball, which means that we can make $x$ from $x_{\text{neighbor}}$ by modifying $n$ components of the sequence following the definition introduced by Sinai et al. (2020).

Figure 14 shows the performance of our method and prior GFN methods in terms of a number of modes. As shown in the figure, our approach outperforms other baselines when the definition of the mode is changed. We also find that when we eliminate a similar sample from the modes, GTB shows promising results among all the other prior GFN methods. Figure 15 also exhibits similar trend.

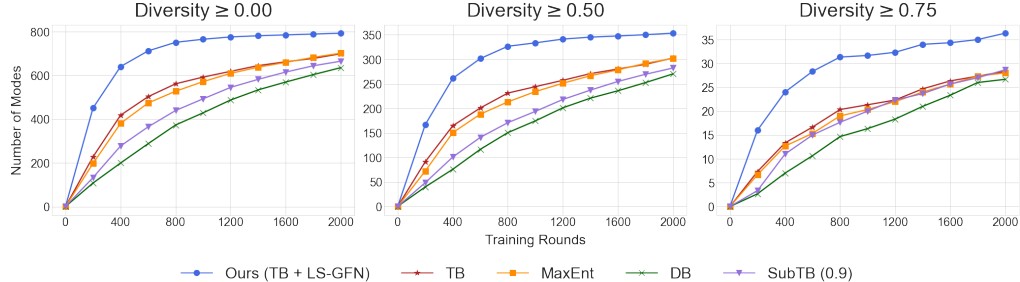

Figure 14: Experiments on several number of modes metrics. Experiments are conducted on QM9. The diversity is measured by 1 - Tanimoto similarity.

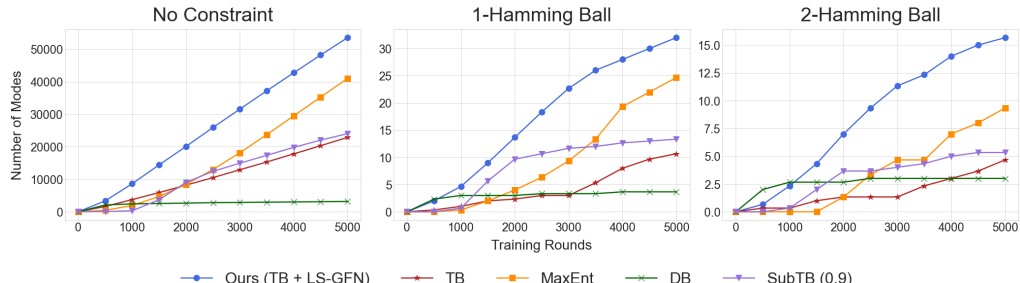

Figure 15: Experiments on several number of modes metrics. Experiments are conducted on L14_RNA1.

### B.5 ABLATION STUDY OF K

We investigate the effect of the number of destruction and reconstruction steps on the performance of our method. For default, we set $K = \lfloor (L+1)/2 \rfloor$, where $L$ is the total length of the object $x$. We conduct an ablations study of $K$ on RNA task. As shown in the Figure 16, we find that when we increase $K$, we can generate more diverse samples while we can achieve higher reward by decreasing $K$. When $k = 4$, we achieve the highest number of modes discovered across training.

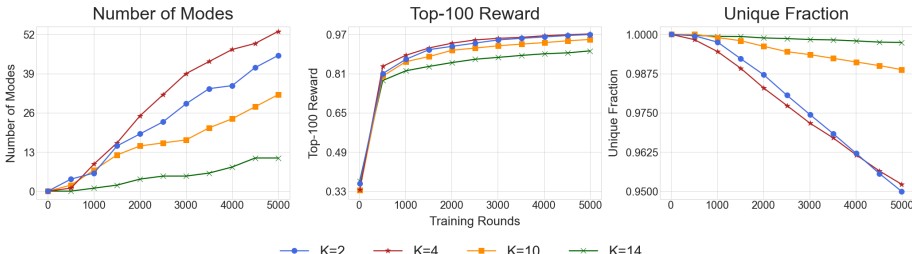

Figure 16: Ablation study on $k$. The average value among 3 independent runs is reported.

### B.6 LOCAL SEARCH ACCEPT RATE EXPERIMENTS

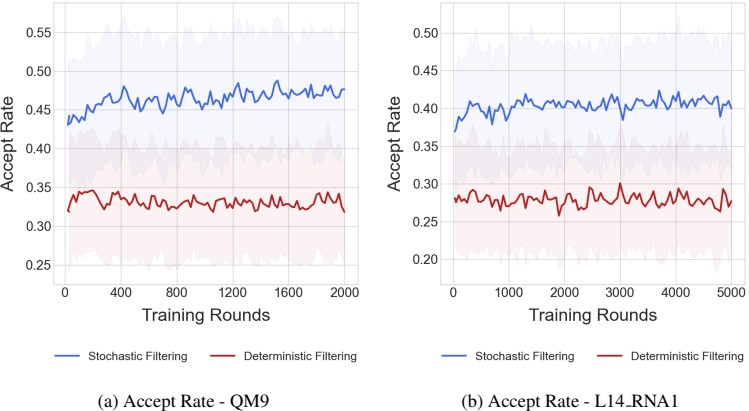

(a) Accept Rate - QM9       (b) Accept Rate - L14_RNA1

Figure 17: Experiments on the local search accept rate of different filtering strategies.

In **Step B** of enhancing the sampled trajectories from $P_F(\tau)$ through a local search guided by $P_B(\tau_{\text{destroy}})$ and $P_F(\tau_{\text{recon}})$, we assess the acceptance rate and decide whether to accept or reject the new suggestion generated by the local search.

Recapping, in deterministic filtering, we accept $\tau'$ with the following probability:

$$A\left(\tau, \tau'\right) = 1_{\{R(\tau') > R(\tau)\}}$$

Additionally, in stochastic filtering, we accept $\tau'$ based on the Metropolis-Hastings acceptance probability:

$$A\left(\tau, \tau'\right) = \min\left[1, \frac{R(\tau')}{R(\tau)} \frac{q(\tau'|\tau)}{q(\tau|\tau')}\right]$$

The acceptance rate, denoted as $A\left(\tau, \tau'\right)$, gauges how effectively local search enhances the performance compared to $P_F(\tau)$. An intriguing experiment involves tracking the acceptance rate during training to observe the dynamic interplay between $P_F(\tau)$ and the local search mechanisms (i.e., $P_B(\tau_{\text{destroy}})$ and $P_F(\tau_{\text{recon}})$). The ideal outcome would manifest as a stable acceptance rate, signifying that as $P_F(\tau)$ evolves efficiently during training, it receives valuable support from the local search, which in turn evolves effectively with the aid of well-trained $P_B(\tau_{\text{destroy}})$ and $P_F(\tau_{\text{recon}})$.

As demonstrated in Figure 17, the acceptance rate remains consistently stable, serving as confirmation that our LS-GFN training maintains stability while evolving both $P_F(\tau)$ and the local search components ($P_B(\tau_{\text{destroy}})$ and $P_F(\tau_{\text{recon}})$) in a mutually supportive manner.

The acceptance rate in deterministic filtering is lower compared to stochastic filtering due to its stricter acceptance criteria. These rates consistently fall below 0.5 in each training iteration, indicating that only a small proportion of successfully refined trajectories contribute significantly to the improvement of the GFlowNet training process.

