# OpenReview forum: "Local Search GFlowNets"
_ICLR.cc/2024/Conference — ICLR 2024 spotlight_

### Official Review · Reviewer_4fxw · 2023-11-01

**Soundness:** 3 good
**Presentation:** 3 good
**Contribution:** 3 good
**Rating:** 8
**Confidence:** 2

**Summary:**

This work proposes an improvement to the GFlowNet training procedure that biases samples towards the high-reward terminal states. The idea is to train a back-tracking model that would go back from the completed flow trajectory in DAG, and then use the forward model to sample the removed part again.
The method boasts impressive performance on biological sequence design when combined with different objectives for training GFlowNets, and achieves the best performance with Trajectory Balance method.

**Strengths:**

- The presentation is clear
- The idea is simple and easy to implement
- The evaluation is comprehensive and showcases the strength of the idea

**Weaknesses:**

- The proposed method adds some computational overhead

**Questions:**

- How is the number of steps K picked for backtracking? If it's fixed, is there a way to pick it automatically?

---

> ### Author Response · Authors · 2023-11-14
>
> Thanks for providing valuable feedback.
>
> ---
>
> **W1: About computational overhead**
>
> LS-GFN does not introduce any additional computational overhead. Below are the wall clock times for each training round in the QM9 task.
>
> |  | $I$ |$M$ | Wall clock time per round|
> | -------- | -------- | -------- | -------- |
> | TB   | 0     | 32     |   1.93 ± 0.13 seconds  |
> | TB + LS-GFN    | 7     | 4     |  1.91 ± 0.07 seconds |
>
> The wall clock time is evaluated with a single RTX 3090 GPU and Intel(R) Xeon(R) Gold 5317 CPU @ 3.00GHz and 256 GB RAM.
>
>
>
>
> This restriction is in place to ensure that the sample complexity remains consistent with that of the baseline GFN methods.
>
> The calculation of sampling complexity follows this formula: $(I+1) \times M$, where $I$ signifies the number of local search iterations, and $M$ denotes the batch size per training round.
>
> | | $I$ |$M$ | Sampling Complexity per Training Round |
> | -------- |-------- |-------- |-------- |
> | GFN        | 0     | 32     | $1 \times 32 = 32$     |
> | LS-GFN         | 7     | 4     |$(7+1) \times 4 = 32$      |
>
> ---
>
> **Q1: How is the number of steps K picked for backtracking? If it's fixed, is there a way to pick it automatically?**
>
> We used $K=\lfloor (L+1) / 2\rfloor$, where $L$ is length of trajecotry. We also provided a hyperparameter analysis in the original manuscript in Appendix B.5. As you mentioned, we can automatically pick $K$ to adaptively balance the wideness of the local search region, which could be a great avenue for future work.

---

> > ### Author Response · Authors · 2023-11-21
> > **Reminder: Author-Reviewer Discussion Period Ending Soon**
> >
> > We wanted to remind you that the author-reviewer discussion period ends in just two days. We value your feedback and kindly request your response to our rebuttal.
> >
> > Your participation will significantly help us enhance our work. If you have any questions or need more information, please don't hesitate to ask.
> >
> > We appreciate your time and look forward to your input.

---

> > > ### Author Response · Authors · 2023-11-22
> > > **Friendly reminder: The Author-Reviewer Discussion Period is concluding in just 24 hours.**
> > >
> > > Dear Reviewer 4fxw,
> > >
> > > Thank you for your role as a reviewer. Since we have a limited 24-hour window for author-reviewer discussion, we kindly ask for your valuable feedback on our rebuttal.
> > >
> > > Best, Authors

---

### Official Review · Reviewer_xEGb · 2023-11-04

**Soundness:** 3 good
**Presentation:** 3 good
**Contribution:** 2 fair
**Rating:** 6
**Confidence:** 4

**Summary:**

The paper proposes Local Search GFlowNet (LS-GFN) for training GFlowNets with local search to enhance the training effectiveness. The proposed algorithm explores the local neighborhood through destruction and reconstruction guided by backward and forward policies, respectively. Extensive experiments demonstrate significant performance improvements in several biochemical tasks.

**Strengths:**

1. The paper introduces an algorithm which combines inter-mode global exploration with intra-mode local exploration in GFlowNets training.
2. The paper is well-written and easy to understand. The proposed algorithm and experimental setup are clearly described.

**Weaknesses:**

1. The paper does not declare the sampling complexity of the proposed method. The local search may require more sampling, which can lead to an unfair comparison.
2. The limitations of the proposed algorithm and potential drawbacks are not discussed in detail.

**Questions:**

1. What about the sampling complexity of local search? Can you conduct experiments under the same sampling complexity or compare your algorithm with an upper bound sampling times?
2. How to decide the local search interaction I? What about the results with different I?
3. As PRT is adopted in LS-GFN training, is PER used for RL methods in te experiments?

---

> ### Author Response · Authors · 2023-11-14
>
> Thanks for the valuable feedback.
>
> ---
>
> **W1, Q1: The paper does not declare the sampling complexity of the proposed method. The local search may require more sampling, which can lead to an unfair comparison.**
>
> Thank you for highlighting this for clarification. All our experiments were conducted under a fair setting, as we used exactly the same number of samples across all experiments. Note this is already outlined in Section 5.3.
>
> The calculation of sampling complexity follows this formula: $(I+1) \times M$, where $I$ stands for the number of local search iterations, and $M$ denotes the batch size per training round.
>
>
>
> | | $I$ |$M$ | Sampling Complexity per Training Rounds |
> | --------  | -------- |-------- |-------- |
> | GFN      | 0     | 32     | $1 \times 32 = 32$     |
> | LS-GFN        | 7     | 4     |$(7+1) \times 4 = 32$      |
>
> ---
>
> **W2: The limitations of the proposed algorithm and potential drawbacks are not discussed in detail.**
>
>
> A limitation of LS-GFN lies in the potential impact of the quality of the backward policy on its performance, particularly when the acceptance rate of the local search becomes excessively low. One immediate remedy is to introduce an exploratory element into the backward policy, utilizing techniques like $\epsilon$-greedy or even employing a uniform distribution to foster exploration within the local search.
>
> A promising avenue for future research could involve fine-tuning backward policy to enhance the acceptance rate of the local search. We plan to incorporate this as a topic in our paper's "Limitations and Further Work" after we get one additional page for the final paper (we now put this into the appendix), and we appreciate your valuable suggestion.
>
> ---
>
> **Q2: How to decide the local search interaction I? What about the results with different I?**
>
> We have conducted an ablation study on the hyperparameter $I$, as now detailed in Appendix B.3.
>
> It's worth noting that across various hyperparameter candidates, namely $I \in \{1, 3, 7, 15, 31\}$, LS-GFN ($I > 0$) consistently outperforms GFN ($I = 0$). This observation suggests that choosing the appropriate hyperparameter $I$ for LS-GFN is relatively straightforward, as it consistently yields superior results.
>
> ---
>
> **Q3: As PRT is adopted in LS-GFN training, is PER used for RL methods in te experiments?**
>
> Taking into consideration your feedback, we implemented reward-prioritized experience replay training in conjunction with the off-policy RL baseline, SQL. It's worth noting that PPO and A2C are on-policy RL methods in which replay training is not directly utilized. Here are new experiment results that compare replay-trained SQL and LS-GFN:
>
>
>
> | | Number of Modes | TopK reward |
> | -------- | -------- | -------- |
> | SQL    | 232 ± 8     | 0.56 ± 0.01     |
> |   SQL (replay trained)   | 235 ± 6     | 0.56 ± 0.01     |
> | TB + LS-GFN   | 793 ± 4     | 0.67 ± 0.00     |

---

> > ### Author Response · Authors · 2023-11-21
> > **Reminder: Author-Reviewer Discussion Period Ending Soon**
> >
> > We wanted to remind you that the author-reviewer discussion period ends in just two days. We value your feedback and kindly request your response to our rebuttal.
> >
> > Your participation will significantly help us enhance our work. If you have any questions or need more information, please don't hesitate to ask.
> >
> > We appreciate your time and look forward to your input.

---

> > > ### Author Response · Authors · 2023-11-22
> > > **Friendly reminder: The Author-Reviewer Discussion Period is concluding in just 24 hours.**
> > >
> > > Dear Reviewer xEGb,
> > >
> > > Thank you for your role as a reviewer. Since we have a limited 24-hour window for author-reviewer discussion, we kindly ask for your valuable feedback on our rebuttal.
> > >
> > > Best, Authors

---

### Official Review · Reviewer_wUkH · 2023-11-06

**Soundness:** 3 good
**Presentation:** 4 excellent
**Contribution:** 3 good
**Rating:** 6
**Confidence:** 3

**Summary:**

- This paper introduces local-search generative flow networks (LS-GFN), a method to improve exploitation in generative flow networks (GFNs).
- The authors argue that, while GFNs explore effectively, their learning is hampered by exploitation — that is, they sample candidate objects that are mostly good but fail to obtain the highest rewards.
- To address this limitation, they propose to refine candidate objects by backtracking
- They validate LS-GFNs on real-world biochemical testbeds, where LS-GFNs significantly outperforms alternatives.
- The paper also includes a well-written introduction to GFNs (Section 3) which covers all the necessary background for the paper.

**Strengths:**

- Novelty: the method is novel as it cleverly takes advantage of both the probabilistic forward and backward policy of GFNs.
- Significance: the proposed methods exhibits excellent empirical results and drastically outperforms the other baselines. I wish there were more of them, but more on that below.
- Clarity: the paper is well written and provides an exceptionally clear and concise introduction to generative flow networks. Moreover the method is quite simple (I think I could replicate the authors’ results) so it’s likely to be adopted by the community.

**Weaknesses:**

- While the method itself is simple and well presented, it remains a little unclear the relative importance of different components. For example, the exposition focuses on the trajectory balanced objective — does this mean LS-GFNs don’t work with different training objectives? The paper also mentions building on top of Shen et al., 2023 which introduces prioritized replay training (PRT) to GFNs, but this method doesn’t appear as a baseline — so how much of the improvements are due to PRT and how much is due to the local search?
- In addition to the hyper-parameter $I$ (number of revisions with local searches), I wished the number of backtracking steps and the acceptance rate were also studied — how much tuning did they require in order to get these good results?
- Wall-clock time is never mentioned — how much overhead does the refining process of LS-GFNs incur? How about in terms of sampled states (instead of training rounds)?
- One baseline I wished were included: given a first trajectory, resample the last K steps and only keeping the best candidate. This is akin to beam search in LLMs or go-explore in RL. Other nice-to-have baselines include top-p and top-k sampling.

**Questions:**

- Why call it “destroy”  since the original trajectory isn’t always discarded? A more intuitive name could be “backtrack”, “rewind”, or anything that doesn’t suggest destruction.
- The top plots in Figure 5 are strange: it looks like some curves go beyond 100% accuracy. Could you either fix them so we can still see the curves on the plot, or explain what is happening?
- How can LS-GFNs recover from a biased backward policy? In other words, assume the forward policy is fine but the backward policy always backtracks to states which yield the same (high reward) candidate objects — how can LS-GFNs overcome this lack of exploration?
- Please confirm that lines 7 and 8 in Algorithm 1 aren’t swapped. If they aren’t (which partially addresses my question above), wouldn’t swapping them and extending $\mathcal{D}$ with $\tau_m$ further improve exploitation? Maybe this should be added as an ablation as well.

---

> ### Author Response · Authors · 2023-11-14
>
> Thanks for your valuable comments.
>
> ---
>
> **W1-1: While the method itself is simple and well presented, it remains a little unclear the relative importance of different components. For example, the exposition focuses on the trajectory-balanced objective — does this mean LS-GFNs don’t work with different training objectives?**
>
>
> LS-GFN is a versatile technique that can be effectively employed for a wide range of GFN objectives. We have already successfully applied LS-GFN to various GFN objectives, including Sub-trajectory balance (SubTB), detailed balance (DB), and maximum entropy GFN (MaxEnt), as demonstrated in Table 1 and Table 2 in our original manuscript.
>
>
> ---
>
> **W1-2: The paper also mentions building on top of Shen et al., 2023 which introduces prioritized replay training (PRT) to GFNs, but this method doesn’t appear as a baseline — so how much of the improvements are due to PRT and how much is due to the local search?**
>
> For a fair comparison, we consistently use "PRT" for training and "SSR" for parameterization in all GFN baselines. Additionally, for direct comparison purposes, we assessed our approach against Shen et al., 2023, denoted as "GTB" in Figure 5 and Figure 6.
>
> It's important to note that Shen et al., 2023 introduced three distinct techniques: "PRT" (pertaining to dataset sampling), "SSR" (concerning GFN parameterization), and "GTB" (associated with the guided trajectory balance objective).
>
> ---
>
> **W2: In addition to the hyper-parameter
>  (number of revisions with local searches), I wished the number of backtracking steps and the acceptance rate were also studied — how much tuning did they require in order to get these good results?**
>
> We've already conducted experiments on hyperparameter $I$ in Appendix B.3, as well as explored the acceptance rate in Appendix B.6.
>
> It's worth noting that across various hyperparameter candidates, namely $I \in \{1, 3, 7, 15, 31\}$, LS-GFN ($I > 0$) consistently outperforms GFN ($I = 0$). This observation suggests that choosing the appropriate hyperparameter $I$ for LS-GFN is relatively straightforward, as it consistently yields superior results.
>
> ---
>
> **W3: Wall-clock time is never mentioned — how much overhead does the refining process of LS-GFNs incur? How about in terms of sampled states (instead of training rounds)?**
>
>
> There's negligible additional wall clock time overhead.
>
> Here are the wall clock times for each training round in the QM9 task. Wall clock time is computed as the mean of three independent seeds. To remove the bias from warmup, we compute the wall clock time per round after 10 initial training rounds. There is no meaningful wall time difference between the two methods:
>
> |  | $I$ |$M$ | Wall clock time per round|
> | -------- | -------- | -------- | -------- |
> | TB   | 0     | 32     |   1.93 ± 0.13 seconds  |
> | TB + LS-GFN    | 7     | 4     |  1.91 ± 0.07 seconds |
>
>
> The wall clock time is evaluated with a single RTX 3090 GPU and Intel(R) Xeon(R) Gold 5317 CPU @ 3.00GHz and 256 GB RAM.
>
>
> The wall clock time remains consistent because we always use the same number of samples per training round ($M \times (I+1) = 32$, $M$: batch size, $I$: number of local searches) for every method across all experiments. This choice is particularly crucial considering that evaluating the reward per sample is a significant bottleneck (in some applications, one could spend a day for reward computation), especially in real-world scenarios like bio and chemical discovery tasks. By keeping the sample complexity uniform, we ensure that the computational overhead remains stable.

---

> > ### Author Response · Authors · 2023-11-14
> >
> > **W4: One baseline I wished was included: given a first trajectory, resample the last K steps, and only keeping the best candidate. This is akin to beam search in LLMs or go-explore in RL. Other nice-to-have baselines include top-p and top-k sampling.**
> >
> > Thank you for recommending baselines that can effectively showcase the LS-GFN method. We update a new section in Appendix B.8, including new experiments and analysis of results. Here is a summarized version of Appendix B.8.
> >
> > We conducted new experiments based on your recommendations. In our evaluation, we compared LS-GFN to the baseline methods you suggested, specifically focusing on the Trajectory Balance (TB) objective function in the QM9 task.
> >
> > In the context of resampling, we initiate a trajectory and subsequently resample the last $K$ steps, repeating this process four times. We concurrently evaluate and select the best trajectory among these resampled trajectories within batches of size $M=8$. Consequently, the total sampling complexity amounts to $8 \times 4 = 32$.
> >
> > In the case of Top K filtering, we generate a total of $M=32$ trajectories and apply a filtering process to identify the Top K  trajectories (Top2 among 32 trajectories), which are then utilized for training purposes.
> >
> >
> > **< QM9 >**
> > | | Number of modes|Top K Reward|
> > | -------- | -------- | -------- |
> > | TB    | 699 ± 14     | 0.66 ± 0.00   |
> > | TB + Resampling     |  778 ± 6   | **0.67 ± 0.00**     |
> > | TB + Top K filtering   | 768 ± 6  | **0.67 ± 0.00**      |
> > | TB + LS-GFN | **793 ± 4**   | **0.67 ± 0.00**   |
> >
> > **< L14_RNA1 >**
> > | | Number of modes|Top K Reward|
> > | -------- | -------- | -------- |
> > | TB    |   6 ± 1  | 0.87 ± 0.01  |
> > | TB + Resampling     |  4 ± 0    | 0.87 ± 0.01   |
> > | TB + Top K filtering   | 4 ± 2  |  0.87 ± 0.01  |
> > | TB + LS-GFN | **18 ± 0**   |  **0.97 ± 0.00** |
> >
> > In the QM9 task, LS-GFN outperforms baselines, having superior performances in terms of number modes. For the RNA task, which contains a larger search space than QM9, an effective search for reward improvement is crucial. The result shows that simple resampling and filtering method is not enough as they are too focused on the exploitation of existing trajectories and are unable to find novel modes effectively.
> >
> > However, LS-GFN harnesses a backward policy for backtracking, resulting in a diverse set of backtracked states that are subsequently reconstructed by the forward policy. This approach has proven to be notably more effective in navigating the extensive RNA search space compared to the resampling method, which merely samples the last $K$ steps without taking advantage of a backward policy.

---

> ### Author Response · Authors · 2023-11-14
> **Responses to the questions**
>
> **Q1: Why call it “destroy” since the original trajectory isn’t always discarded? A more intuitive name could be “backtrack,” “rewind,” or anything that doesn’t suggest destruction.**
>
> We are in agreement on this matter. We have updated our manuscript to incorporate the term "backtrack" instead of "destroy."
>
> ---
>
> **Q2: The top plots in Figure 5 are strange: it looks like some curves go beyond 100% accuracy. Could you either fix them so we can still see the curves on the plot or explain what is happening?**
>
> Please note that Figure 5 is plotted accurately in accordance with the baseline from the paper by Shen et al., 2023, where we have set our accuracy metrics following their guidelines:
>
>
> $\text{Acc}(p(x;\theta)) = 100 \times \text{min}(\frac{E_{p(x;\theta)}[R(x)]}{E_{p^{*}(x)}[R(x)]},1)$
>
>
> This provides an upper bound of 100% accuracy. In response to your suggestion, we have included a graph illustrating accuracy beyond the 100% mark in Appendix B.7.
>
> ---
>
> **Q3: How can LS-GFNs recover from a biased backward policy? In other words, assume the forward policy is fine but the backward policy always backtracks to states that yield the same (high reward) candidate objects — how can LS-GFNs overcome this lack of exploration?**
>
> Biased backward policies can indeed pose challenges. In such scenarios, we can employ an $\epsilon$-greedy method to introduce exploratory back-sampling into the biased backward policy. It's worth noting that LS-GFN performs exceptionally well in MaxEnt GFN, even when the backward policy is set to a uniform distribution (i.e., $\epsilon = 1$).
>
> Building upon your valuable feedback, one potential avenue for future research could involve fine-tuning the backward policy to maximize the local search acceptance rate. This approach would aim to ensure that the backward policy identifies diverse intermediate states that lead to improved rewards without exhibiting a bias towards yielding the same candidate objects repeatedly. We appreciate your insightful comment.
>
> ---
>
> **Q4: Please confirm that lines 7 and 8 in Algorithm 1 aren’t swapped. If they aren’t (which partially addresses my question above), wouldn’t swapping them and extending them further improve exploitation? Maybe this should be added as an ablation as well.**
>
> No swapping is involved here. We update every sample to the dataset, irrespective of whether it is accepted or not. The rationale behind this approach is that every sample where the reward is calculated is considered valuable. During the refinement phase, our primary objective is to explore the local region in order to discover high-reward samples rather than determining whether the found sample should be used in training.
>
> In the training process, we employ PRT to sample trajectories in proportion to their rewards from the dataset. Consequently, accepted samples have a higher likelihood of being included during training.
>
> Regarding your suggestion to swap lines 7 and 8, it would entail updating the data only with accepted samples. However, as you pointed out, while this could enhance exploitation, it might excessively hinder exploration. To shed more light on this, here are the new results of our ablation study:
>
>
>
> | Column 1 | Number of modes| TopK reward |
> | -------- | -------- | -------- |
> | Swapped version     | 754 ± 6    | 0.66 ± 0.00    |
> | Original LS-GFN    | 793 ± 4     | 0.67 ± 0.00    |
>
> Based on the above results, the original LS-GFN gives higher performances than the swapped version.
>
> ---
>
> **References**
>
> Max W Shen, Emmanuel Bengio, Ehsan Hajiramezanali, Andreas Loukas, Kyunghyun Cho, and
> Tommaso Biancalani. Towards understanding and improving GFlowNet training. In International
> Conference on Machine Learning, pp. 30956–30975. PMLR, 2023

---

> > ### Author Response · Authors · 2023-11-21
> > **Reminder: Author-Reviewer Discussion Period Ending Soon**
> >
> > We wanted to remind you that the author-reviewer discussion period ends in just two days. We value your feedback and kindly request your response to our rebuttal.
> >
> > Your participation will significantly help us enhance our work. If you have any questions or need more information, please don't hesitate to ask.
> >
> > We appreciate your time and look forward to your input.

---

> > > ### Author Response · Authors · 2023-11-22
> > > **Friendly reminder: The Author-Reviewer Discussion Period is concluding in just 24 hours.**
> > >
> > > Dear Reviewer wUkH,
> > >
> > > Thank you for your role as a reviewer. Since we have a limited 24-hour window for author-reviewer discussion, we kindly ask for your valuable feedback on our rebuttal.
> > >
> > > Best,
> > > Authors

---

### Author Response · Authors · 2023-11-14
**Overall Responses**

We would like to express our sincere gratitude to all the reviewers for their valuable comments and feedback. In the following sections, we provide a summary of our revisions and responses to each reviewer's feedback.

**Paper revision**

- In response to feedback from Reviewer wUkH, we have replaced the term "destroy" with "backtrack."

- We have incorporated Reviewer xEGb's suggestion and feedback from Reviewer wUkH by updating the "Limitations and Further Work" section, which is now located in Section 5.

- In line with Reviewer wUkH's feedback, we have made revisions to Appendix B.7, specifically focusing on plotting the accuracy of our sampling method to illustrate accuracy curves extending beyond 100%.

- We updated Appendix B.8 to incorporate the suggestions made by Reviewer wUkH regarding the inclusion of an additional baseline involving resampling and a filtering strategy.

We have highlighted every revised section in purple.

**Summary of Additional Experiments**

We have conducted these additional experiments in response to the raised concerns.

- **Computational Overhead (question raised by all reviewers)**: We assessed the computational overhead of LS-GFN and found that it does not introduce any additional computational load when compared to GFN baselines.

- **Additional Baseline of Resampling and Filtering Strategy (suggested by Reviewer wUkH)**: LS-GFN demonstrates superior performance when compared to these baselines. This is also updated in Appendix B.8.

- **Swapping Lines 7 and 8 in the Pseudo Code (suggested by Reviewer wUkH)**: We conducted experiments to compare the original LS-GFN algorithm with a swapped version, and the original algorithm performed better.

- **Comparison with Replay Training Off-Policy RL Baseline (suggested by Reviewer xEGb)**: We conducted replay training for soft-Q learning and demonstrated that LS-GFN significantly outperforms the baseline.

---

### Meta-Review · Area_Chair_FRwn · 2023-12-10

**Metareview:**

This pape paper proposes Local Search GFlowNet (LS-GFN) for training GFlowNets with local search to enhance the training effectiveness. Reviewers appreciated the paper's novelty, significance, and clarity. The paper received three reviews, all of which rated the paper as an accept and one of which advocated more forcefully. Given that all reviews received thorough replies and that the reviewers did not take the opportunity to push back, the paper is clearly slated for acceptance.

**Justification For Why Not Higher Score:**

The most thorough reviews rate the paper as above the cutoff for acceptance, the stronger positive vote lacks sufficient detail to justify.

**Justification For Why Not Lower Score:**

The paper has received unanimous votes for acceptance and appears to offer a valuable contribution for an emerging topic of interest to the community.

---

### Decision · Program_Chairs · 2024-01-16

Accept (spotlight)